# Solving Nash Equilibrium Scalably via Deep-Learning-Augmented Iterative Algorithms

## Abstract

Computing the Nash Equilibrium (NE) is a fundamental yet computationally challenging problem in game theory. Although recent approaches have incorporated deep learning techniques to tackle this intractability, most of them still struggle with scalability when the number of players increases, due to the exponential growth of computational cost. Inspired by the efficiency of classical learning dynamics methods, we propose a deep learning-augmented Nash equilibrium solver, named Deep Iterative Nash Equilibrium Solver (DINES), based on a novel framework that integrates deep learning into iterative algorithms to solve Nash Equilibria more efficiently. Our approach effectively reduces time complexity to a polynomial level and mitigates the curse of dimensionality by leveraging query-based access to utility functions rather than requiring the full utility matrix. Experimental results demonstrate that our approach achieves better or comparable approximation accuracy compared to existing methods, while significantly reducing computational expense. This advantage is highlighted in large-scale sparse games, which is previously intractable for most existing deep-learning-based methods.

## 1 Introduction

The problem of computing Nash equilibria (NE) is one of the most fundamental challenges in game theory, and has profound implications in artificial intelligence fields, including multi-agent systems, reinforcement learning, and strategic decision-making. Nash equilibria provide a way to predict and understand how multiple rational agents interact and make decisions in competitive or cooperative environments, making it highly relevant for applications ranging from autonomous systems to multi-agent reinforcement learning. The computation of NE gains increasing importance as AI systems increasingly operate in complex, multi-agent settings, such as autonomous driving, market simulations, and resource allocation.

Despite its importance, finding NE in general games faces significant computational challenges, and over the past few decades, extensive research has focused on developing efficient algorithms to address this problem. It is known that the computation complexity of this problem is PPAD-complete (Daskalakis et al., 2006; Chen & Deng, 2006), meaning that any algorithm that computes NE exactly must take exponential time, unless PPAD = P. To circumvent this complexity, polynomial-time algorithms for approximate Nash equilibria have been developed, but improving approximation guarantees remains challenging. Moreover, these algorithms are mostly restricted to two-player games, and do not easily scale to multi-player settings. On the other hand, heuristic methods, especially the class of learning dynamics like Fictitious Play (Brown, 1951) and Regret Matching (Hart & Mas-Colell, 2000), are effectively applied in various practical scenarios, including decentralized environments. Despite their practical success, these methods lack guarantees of convergence to NE, and may fail to converge in worst-case situations. This divide between theoretical algorithms and heuristic methods reflects the constant conflict between computational efficiency and solution quality when computing Nash equilibria.

Inspired by advances in deep learning, recent works have developed new approaches to solve Nash equilibrium by leveraging neural networks, mitigating the computational limitations of traditional solvers. For example, Duan et al. (2023a) firstly proposed to approximate equilibrium strategies with a multilayer perceptron (MLP), trained based on approximation error. Goktas et al. (2024) introduced a generative adversarial network (GAN)-based solver for games with continuous strategy

spaces. Liu et al. (2024) employed attention mechanism over joint actions to compute equilibria with improved accuracy and equivariant property.

However, these methods still face significant challenges. Firstly, they are computationally costly due to the large model size required to achieve reasonable approximation of the equilibrium. Secondly, they struggle with the exponential growth in the size of the utility matrix as the number of players increases, because their model take as input the whole utility matrix, which leads to scalability issues. Moreover, due to the black-box nature of neural networks, these methods lack the interpretability and generalization ability compared with classical algorithms.

In this paper, we propose a deep learning-augmented Nash equilibrium solver, named **Deep Iterative Nash Equilibrium Solver (DINES)**, which combines the strengths of classical learning dynamics with modern deep learning techniques. Our approach replaces the iterative update rules in learning dynamics methods with optimizable deep neural networks, leveraging player-wise and action-wise attention mechanisms to balance expressive power and computational efficiency in the decision of strategies. By optimizing the solver's convergence rate and reducing the complexity of utility access, we achieve significant improvements over existing methods in terms of both speed and scalability.

Our contributions are threefold:

1. We introduce a general framework of learning-augmented iterative NE solver, and propose Deep Iterative Nash Equilibrium Solver (DINES) based on this framework. This framework retains the structural advantages of classical learning dynamics, while leveraging the expressive power of deep learning to enhance convergence performance. We design the model architecture of DINES by incorporating efficient attention-based neural networks, achieving a polynomial computational complexity in the number of players and actions.

2. We demonstrate through experimental results that our method achieves equilibrium approximations superior to or comparable with those of state-of-the-art deep learning methods, while significantly reducing computational costs. Compared to classical learning dynamics, our approach achieves notable improvements in both accuracy and convergence.

3. We conduct further experiments to demeonstrate the scalability and adaptability of our approach in large-scale multi-player succinct games, such as polymatrix games, which have been intractable for most existing methods due to the curse of dimensionality.

## 2 LITERATURE REVIEW

Extensive research has focused on developing a wide range of methods to either solve or approximate Nash equilibria, broadly categorized into three major approaches: theoretical algorithms, heuristic algorithms, and deep-learning approachs. In this section, we review these approaches, summarizing their strengths and limitations. We refer the reader to a recent survey (Li et al., 2024) for a detailed review.

**Theoretical algorithms.** It is well-known that the Nash equilibrium in zero-sum two-player games can be solved efficiently through linear programming, but the problem become PPAD-complete in general-sum games (Daskalakis et al., 2006; Chen & Deng, 2006). Various algorithms have been designed to solve Nash equilibrium in general normal-form games exactly or near-exactly (Lipton et al., 2003; Kontogiannis & Spirakis, 2007; Lemke & Howson, 1964; Laan et al., 1987; Govindan & Wilson, 2003; Porter et al., 2008). All these methods take super-polynomial running time due to the PPAD-complete computational complexity. To circumvent this intractability, polynomial time algorithms for approximate Nash equilibrium are designed (Kontogiannis et al., 2009; Daskalakis et al., 2009; 2007; Bosse et al., 2010; Tsaknakis & Spirakis, 2008; Deligkas et al., 2023). However, this line of research faces significant challenge in improving the approximation guarantees: The current best algorithm by Deligkas et al. (2023) guarantees a $1/3$-approximation, offering only a slight improvement over the $0.3393$-approximation achieved by Tsaknakis & Spirakis (2008) decades ago. The applicability of these approximate algorithms in practical scenarios is limited for two main reasons: firstly the guaranteed approximation ratio remains too large to be meaningful, and secondly they are restricted to two-player bimatrix games, without easy generalization to multi-player games.

**Heuristic algorithms and learning dynamics.** Heuristic algorithms, especially in the form of learning dynamics, have emerged as practical solutions for computing Nash equilibria in games in-

volving multiple players. Based on the heuristic idea that players can reach the equilibrium through repeated interactions, in these dynamics, agents adapt their strategies based on observed outcomes in each round, forming a iterative learning process. The class of learning dynamics includes fictitious play (Brown, 1951), best response dynamics (Cournot, 1838), and no-regret learning algorithms such as regret matching (Hart & Mas-Colell, 2000), Hedge (Auer et al., 1995), and multiplicative weight update methods (Arora et al., 2012). The main advantage of these methods lies in their scalability owing to the simplicity and efficiency of the updating rules. Additionally, many learning dynamics can operate in a decentralized manner, requiring no communication between agents beyond the strategic interaction, which makes these methods particularly suitable for distributed scenarios. However, learning dynamics in general have no theoretical guarantees for convergence rate to NE, and can even fail to converge in worst cases. Nonetheless, a wide range of weaker convergence results have been derived. In specific cases like two-player zero-sum games and potential games, fictitious play is known to converge in terms of the empirical distribution of actions over time (Robinson, 1951; Monderer & Shapley, 1996). For general games, it is proved that any no-regret dynamics converges to coarse correlated equilibrium (CCE), while any no-swap-regret dynamics converges to correlated equilibrium (CE) (Cesa-Bianchi & Lugosi, 2006; Blum & Mansour, 2005).

**Deep-learning approaches.** To avoid the theoretical barriers and pursue better accuracy and efficiency in practical scenarios, a series of deep-learning-based methods have been developed to solve the Nash equilibrium. Duan et al. (2023a) firstly proposes to approximate Nash equilibrium with neural networks, and derive generalization bound and agnostic PAC learnability results. Marris et al. (2022) proposes a deep-learning framework for equilibrium solving and selection, mainly focusing on CE and CCE. Goktas et al. (2024) utilizes generative adversarial learning to solve equilibria in general game-theoretic settings with continuous strategy spaces. Liu et al. (2024) is the most related to our work. They achieve the state-of-the-art approximation accuracy through attention operations with equivariant property. However, their proposed model is computationally expensive due to the self-attention operation on all $T^N$ joint actions, resulting in a $\Theta(T^{2N})$ time complexity, hindering its application to scenarios with larger number of players. This curse of dimensionality is also faced by most existing methods due to direct access to the whole utility table. While our work also utilizes the attention mechanism, we effectively reduce the computational complexity through query access of utilities and decomposed attention operations, obtaining significantly improved efficiency and scalability.

## 3 GAME-THEORETIC PRELIMINARIES

### 3.1 NORMAL-FORM GAME

A normal-form game $G$ is specified by the tuple $(N, T, (G_p(\cdot))_{p \in [N]})$, indicating that there are $N$ players indexed as $[N] = \{1, \cdots, N\}$, with each player $p$ having $T$ actions[1] $a_p^1 \ldots a_p^T$ respectively. We denote $\mathcal{A}_p = \{a_p^1 \ldots a_p^T\}$ as the action space of player $p$ and $\mathcal{A} = \mathcal{A}_1 \times \cdots \times \mathcal{A}_N$ as the joint action space of all players. When each player plays an action $a_p \in \mathcal{A}_p$, a joint action $\boldsymbol{a} = (a_1, \ldots, a_N) \in \mathcal{A}$ is formed, and each player $p$ receives a utility of $G_p(a)$, where $G_p : \mathcal{A} \to \mathbb{R}$ is called the utility function of player $p$.

A mixed strategy of player $p$ is defined as a distribution on $\mathcal{A}_p$, represented by a vector $\boldsymbol{x}_p = (x_{p,1}, \cdots, x_{p,T}) \in \Delta_T$. Here $\Delta_T := \{(x_1, \cdots, x_T) \in [0,1]^T : \sum_{j=1}^T x_j = 1\}$ is the standard $(T-1)$-dimensional simplex. We write $a_p \sim \boldsymbol{x}_p$ when player $p$ follows the mixed strategy $\boldsymbol{x}_p$, meaning that she chooses action $a_p^j$ with probability $x_{p,j}$ for each $j \in [T]$. Given a mixed strategy profile $\boldsymbol{x} = (\boldsymbol{x}_1, \cdots, \boldsymbol{x}_N)$, with a slight abuse of notation, we denote player $p$'s expected utility under $\boldsymbol{x}$ by

$$G_p(\boldsymbol{x}) := \mathbb{E}_{a_i \sim \boldsymbol{x}_i, \, \forall i \in [N]}[G_p(\boldsymbol{a})],$$

and denote player $p$'s utility playing each action $a_p^j$ by

$$G_p(a_p^j, \boldsymbol{x}_{-p}) := \mathbb{E}_{a_i \sim \boldsymbol{x}_i, \, \forall i \neq p}[G_p(a_p^j, \boldsymbol{a}_{-p})].$$

---

[1]For convenience and without loss of generality, we assume all players have the same number of actions throughout the paper.

**Definition 1** (Nash Equilibrium). In a normal-form game $G$, a mixed strategy profile $\boldsymbol{x} = (\boldsymbol{x}_1, \cdots, \boldsymbol{x}_N)$ is a Nash equilibrium, if for all $p \in [N]$, it holds that

$$G_p(\boldsymbol{x}) \geq \max_{j \in [T]} G_p(a_p^j, \boldsymbol{x}_{-p}).$$

For any $\epsilon > 0$, a mixed strategy profile $\boldsymbol{x}$ is an $\epsilon$-approximate Nash equilibrium, if for all $p \in [N]$, it holds that

$$G_p(\boldsymbol{x}) \geq \max_{j \in [T]} G_p(a_p^j, \boldsymbol{x}_{-p}) - \epsilon.$$

**Definition 2** (Nash Approximation). In a normal-form game $G$, for any mixed strategy profile $\boldsymbol{x} = (\boldsymbol{x}_1, \cdots, \boldsymbol{x}_N)$, we define its Nash approximation as

$$\mathrm{NashAppr}(\boldsymbol{x}) := \max_{p \in [N]} \max_{j \in [T]} (G_p(a_p^j, \boldsymbol{x}_{-p}) - G_p(\boldsymbol{x})).$$

One can see that $\boldsymbol{x}$ is an $\epsilon$-approximate Nash equilibrium (or an Nash equilibrium, respectively) if and only if $\mathrm{NashAppr}(\boldsymbol{x}) \leq \epsilon$ (or $\mathrm{NashAppr}(\boldsymbol{x}) = 0$, respectively).

### 3.2 GAME REPRESENTATION

From a computational perspective, the representation of the utility functions $G_p(\cdot)$ in a normal-form game determines how the algorithm can access it. Below we introduce several kinds of representations:

**Tabular Game** In a tabular game, the utility functions are explicitly represented as a $N$-dimensional payoff table that lists the players' utility values under every joint action.

**Polymatrix Game** A polymatrix game can be decomposed into bimatrix games (i.e., two-player tabular game) between every pair of players. The utility of a player is obtained by summing up her utility in all the $N - 1$ bimatrix games involving her. The utility functions are represented by $N(N - 1)$ matrices indicating the payoff tables of all these bimatrix games.

**Succinct Game** A succinct game refers to any game with a succinct representation of the utility functions that is feasible under large number of players and actions, including the polymatrix game. Most generally, each utility function $G_p(\cdot)$ is represented by a polynomial-sized circuit that can be evaluated on the joint action profiles.

When a game has specific structures such as decomposable or sparse interactions, representing it as a succinct game can significantly reduce the size of the representation compared to the tabular form, bringing advantages in computational efficiency.

### 3.3 PERMUTATION EQUIVARIANCE

The concept of Nash equilibrium possesses a natural property of permutation equivariance: When the players and actions are reordered, the Nash equilibria are reordered in the same way. Consequently, it is reasonable for a Nash equilibrium solver to also satisfy permutation equivariance. It is also suggested in Duan et al. (2023b) that permutation equivariance can improve the solver's generalization ability. We formally define this property as follows.

**Definition 3** (Permutation Equivariance). Let $\mathcal{G}_{N,T}$ denote the space of all normal-form games $G$ with given $N$ and $T$. An isomorphism $\mathrm{Iso}_\rho$ on $\mathcal{G}_{N,T}$ is specified by $\rho = (\pi, \tau_1, \ldots, \tau_N)$, where $\pi$ is a permutation on the player set $[N]$, and each $\tau_p$ is a permutation on the action set $\mathcal{A}_p$. Such a isomorphism maps a game $G \in \mathcal{G}_{N,T}$ to a new game $G' = \mathrm{Iso}_\rho(G)$ where the utility functions are given by

$$G'_{\pi(p)}(\tau_{\pi(1)}(a_{\pi(1)}), \ldots, \tau_{\pi(N)}(a_{\pi(N)})) = G_p(a_1, \ldots, a_N).$$

Consider a mapping $\phi = (\phi_{p,j})_{p \in [N], j \in [T]} : \mathcal{G}_{N,T} \to \Omega^{[N] \times [T]}$, which produces an output $\phi_{p,j}(G) \in \Omega$ for each action $a_p^j$. Here $\Omega$ denotes the output space. We define $\phi$ to be permutation equivariant, if for any isomorphism $\rho = (\pi, \tau_1, \ldots, \tau_N)$, it holds for all $G \in \mathcal{G}_{N,T}$ and $p \in [N]$, $j \in [T]$ that

$$\phi_{\pi(p), \tau_{\pi(p)}}(\mathrm{Iso}_\rho(G)) = \phi_{p,j}(G).$$

Similarly, consider a mapping $\phi = (\phi_p)_{p \in [N]} : \mathcal{G}_{N,T} \to \Omega^{[N]}$, which produces an output $\phi_p(G) \in \Omega$ for each player $p$, with $\Omega$ denoting the output space. We define $\phi$ to be permutation equivariant if for any isomorphism $\rho = (\pi, \tau_1, \ldots, \tau_N)$, it holds for all $G \in \mathcal{G}_{N,T}$ and $p \in [N], j \in [T]$ that

$$\phi_{\pi(p)}(\text{Iso}_\rho(G)) = \phi_p(G).$$

We say a deterministic algorithm is permutation equivariant, if it can be viewed as a permutation equivariant mapping from the game to a strategy profile. For a randomized algorithm, we can similarly define it to be permutation equivariant if the output distribution is permutation equivariant.

## 4 DEEP ITERATIVE NASH EQUILIBRIUM SOLVER (DINES)

### 4.1 MOTIVATION AND GENERAL FRAMEWORK

---
**Algorithm 1** Common Structure of Learning Dynamics

---
**Require:** number of players $n$, number of actions $T$, utility functions $G_1(\cdot), \cdots, G_n(\cdot)$.

Initialize inner states: $\boldsymbol{y}^{(0)} = (y_{p,j}^{(0)})_{p \in [N], j \in [T]}$.

$k \leftarrow 0$.

**repeat**

    $k \leftarrow k + 1$.

    Generate mixed strategy profile: $\boldsymbol{x}_p^{(k)} \leftarrow \phi(y_{p,:}^{(k-1)}; k), \forall p \in [N]$.

    Query utility of each action: $u_{p,j}^{(k)} \leftarrow G_p(a_p^j, \boldsymbol{x}_{-p}^{(k)}), \forall p \in [N], j \in [T]$.

    Update inner states: $y_{p,:}^{(k)} \leftarrow \psi(y_{p,:}^{(k-1)}, (u_{p,j}^{(k)})_{j \in [T]}; k), \forall p \in [N]$.

**until** reaching maximum iteration or convergence condition

Generate output mixed strategy profile: $\hat{\boldsymbol{x}}_p \leftarrow \hat{\phi}(y_{p,:}^{(k)}), \forall p \in [N]$.

**return** $\hat{\boldsymbol{x}} = (\hat{\boldsymbol{x}}_1, \cdots, \hat{\boldsymbol{x}}_N)$.

---

In scenarios with large number of players and actions, the growth in computational cost becomes the main challenge for most existing algorithms in solving the Nash equilibrium. Learning dynamics methods are often favored due to their efficiency and adaptability, largely owing to their iterative nature. This motivates us to combine the structural advantage of learning dynamics with the expressive power of deep learning models.

Most learning dynamics methods for NE computation follow a common iterative structural framework, as presented in Algorithm 1. Under this framework, players interact iteratively, each maintaining a internal state, which is typically a vector, such as the player's current evaluation of the actions. In each iteration, each player first determines their current strategy (either pure or mixed) based on their states, as represented by the mapping $\phi$ in the algorithm. Subsequently, players calculate the utility of their each action given the current strategy profile of other players. At the end of each iteration, players update their internal states based on these utilities, as represented by the mapping $\psi$. The iteration process continues until a convergence condition is satisfied or a predefined number of iterations is reached. Finally, the output strategy profile is generated and output according to the players' inner states at the final round, as represented by the mapping $\hat{\phi}$.

Inspired by this common structure of learning dynamics, we propose a framework of deep-learning augmented iterative algorithms for solving the Nash equilibrium, as presented in Algorithm 2. The framework unfolds the iteration process into a fixed number of $K$ rounds, and implement the functions $\phi$ and $\psi$ in each round as deep neural networks $\Phi$ and $\Psi$, where $\Phi$ is for generating the mixed strategies, and $\Psi$ is for updating the inner states. They are parametered by $\theta_\Phi^{(1)}, \cdots, \theta_\Phi^{(K)}$ and $\theta_\Psi^{(1)}, \cdots, \theta_\Psi^{(K)}$, respectively. Additionally, $\hat{\phi}$ is also computed by $\Phi$ with parameter $\hat{\theta}_\Phi$. Different from Algorithm 1, here we allow a centralized update of all inner states in this framework to enhance expressive power. Assuming the game instance is generated from a certain distribution, all these parameters can be trained with data samples. In the next subsection, we will describe the architecture of the neural networks $\Phi$ and $\Psi$, which constitutes our proposed DINES model.

This framework brings the following advantages:

---

**Algorithm 2** A Framework of Deep-learning Augmented Iterative Algorithm

---

**Require:** number of players $n$, number of actions $T$, number of rounds $K$, initial embeddings $\boldsymbol{y}^{(0)}$,
   utility functions $G_1(\cdot), \cdots, G_n(\cdot)$, weight parameters $\theta_\Phi^{(1)}, \cdots, \theta_\Phi^{(K)}, \hat{\theta}_\Phi$ and $\theta_\Psi^{(1)}, \cdots, \theta_\Psi^{(K)}$.
   Initialize inner states: $\boldsymbol{y}^{(0)} = (y_p^{(0)})_{p \in [N]}$.
   **for** $k \leftarrow 1, \cdots, K$ **do**
      Generate mixed strategy profile: $\boldsymbol{x}_p^{(k)} \leftarrow \Phi(y_p^{(k-1)}; \theta_\Phi^{(k)}), \forall p \in [N]$.
      Query utility of each action: $u_{p,j}^{(k)} \leftarrow G_p(a_p^j, \boldsymbol{x}_{-p}^{(k)}), \forall p \in [N], j \in [T]$.
      Update inner states: $\boldsymbol{y}^{(k)} \leftarrow \Psi(\boldsymbol{y}^{(k-1)}, (u_{p,j}^{(k)})_{p \in [N], j \in [T]}; \theta_\Psi^{(k)})$
   **end for**
   Generate output strategy profile: $\hat{\boldsymbol{x}}_p \leftarrow \Phi(y_p^{(k)}; \hat{\theta}_\Phi), \forall p \in [N]$.
   **return** $\hat{\boldsymbol{x}} = (\hat{\boldsymbol{x}}_1, \cdots, \hat{\boldsymbol{x}}_N)$.

---

- Computational efficiency and scalability. Most existing algorithms for Nash equilibrium, aside from the class of learning dynamics methods, require a direct access to the whole utility matrix, especially deep-learning approaches. Therefore, they unavoidably suffers from the curse of dimensionality, as the size of the utility matrix grows exponentially in the number of players. In contrast, under our framework, the utility is accessed through queries to the utility functions. Throughout the execution of our framework, only $KnT$ utility queries are made, where $nT$ queries happens in each round, and the total number of rounds $K$ is typically not too large. Moreover, as long as the game has a succinct representation, each query of utility function can be efficiently computed, or at least unbiasedly estimated. This establishes the polynomial computational cost of our framework, making it applicable to games involving a much larger number of players.

- Improved convergence. Due to the expressive power and data-driven optimization, the convergence of iterative process can be largely accelerated compared to traditional learning dynamics.

- Permutation equivariance. As long as the functions $\Phi$ and $\Psi$ satisfy permutation equivariance, and the inner states are initialized identically (or from identical distribution), the resulted algorithm will preserve permutation equivariance.

- Structural explanability. As the expressive power of deep learning models can easily represent the updating rules in classical learning dynamics, the resulted algorithm can be interpreted as a strengthened version of learning dynamics, inheriting the heuristic explanability.

## 4.2 MODEL ARCHITECTURE

In this subsection, we describe our proposed model architecture of Deep Iterative Nash Equilibrium Solver (DINES), building on the general framework introduced in the last subsection. Our design focuses on achieving two key properties: computational efficiency and permutation equivariance, which motivate the use of player-wise and action-wise attention mechanisms.

### STRUCTURE OF INNER STATES

In DINES, the inner state $y_p^{(k)}$ of each player is separated into three parts: a record of the latest chosen mixed strategy $\boldsymbol{x}_p^{(k)}$, the player embedding $\beta_p^{(k)} \in \mathbb{R}^D$, and the action embeddings $\alpha_{p,j}^{(k)} \in \mathbb{R}^D$ for each $j \in [T]$. Here $D$ is a hyper-parameter representing the dimension of the embedding space. Recording the last mixed strategy allows a residual updating of strategies, which is described later. The player embedding $\beta_p^{(k)}$ of player $p$ captures her overall characteristics and her global information acquired from the centralized updates. The action embedding $\alpha_{p,j}^{(k)}$ represent the player's fine-grained evaluation and belief about each available action $a_p^j$, affecting her strategic decision in each round.

### INITIALIZATION

Instead of using a deterministic initialization value, we initialize the inner states $\boldsymbol{y}^{(0)}$ randomly for DINES. Each embedding $\beta_p^{(0)}$ and $\alpha_{p,j}^{(0)}$ is drawn independently from the $D$-dimensional standard Gaussian distribution $\mathcal{N}^D$. The initial recorded mixed strategy is generated by firstly drawing $\tilde{x}_{p,j}$ iid from $U[0,1]$, and then normalizing to $x_{p,j}^{(0)} = \frac{\tilde{x}_{p,j}}{\sum_{j' \in [T]} \tilde{x}_{p,j}}$.

Notably, this randomized initialization allows DINES to find asymmetric equilibria in games with symmetric players or symmetric actions, while retaining the permutation equivariance. In comparison, if a algorithm is both deterministic and permutation equivariant, then its output must be symmetric for symmetric players (or symmetric actions). Although symmetric equilibria are guaranteed to exist in symmetric games, Duan et al. (2023b) have shown that the restriction to symmetric output can result in arbitrarily large loss in social welfare compared with the asymmetric equilibria. By identically and independently initializing the embeddings of players and actions, our model naturally breaks the symmetry, while the output distribution still guarantees permutation equivariance.

### GENERATION OF MIXED STRATEGY

The function $\Phi$ for generating the mixed strategies from the inner states is simply implemented as a feed forward network (FFN) which maps each action embedding to a logit, which is then turned into a distribution on action space by the softmax operator. This can be represented as follows:

$$\boldsymbol{x}_p^{(k)} = \Phi(y_p^{(k-1)}; \theta_\Phi^{(k)}) = \text{softmax}(\text{FFN}(\alpha_{p,1}^{(k-1)}; \theta_\Phi^{(k)}), \cdots, \text{FFN}(\alpha_{p,T}^{(k-1)}; \theta_\Phi^{(k)})).$$

### UPDATING OF EMBEDDINGS

The updating of all embeddings in each round, i.e. the function $\Psi$, is decomposed into four phases, mainly consisting of attention operations. For the ease of notation, here we omit the superscript of $^{(k)}$ on intermediate embeddings. We also omit the weight parameters and feed forward networks in attention operations.

**Action-wise self-attention.** The first phase updates the action embeddings for each player individually, given the result of utility quries $u_{p,j}^{(k)}$. This is done through a self-attention:

$$(\alpha_{p,j}')_{j \in [T]} = \text{selfAttention}\left(\left(\text{concat}(\alpha_{p,j}^{(k-1)}, u_{p,j}^{(k)})\right)_{j \in [T]}\right).$$

**Player-action attention.** In the second phase, we update the player embedding for each player individually, aggregating the updated information of her actions through a single-query attention:

$$\beta_p' = \text{Attention}\left(\beta_p^{(k-1)}, \left(\text{concat}(\beta_p^{(k-1)}, \alpha_{p,j}')\right)_{j \in [T]}\right).$$

**Player-wise self-attention.** In the third phase, we update the player embeddings globally through a player-wise self-attention, obtaining the final player embeddings for this round:

$$(\beta_p^{(k)})_{p \in [N]} = \text{selfAttention}\left((\beta_p')_{p \in [N]}\right).$$

**Action-player update.** In the fourth phase, we update the action embeddings with, obtaining the final action embeddings for this round:

$$\alpha_{p,j}^{(k)} = \text{FFN}\left(\text{concat}(\alpha_{p,j}', \beta_p^{(k)})\right).$$

Notably, since all operations in the updating of embeddings satisfies permutation equivariance, the output distribution of DINES is also permutation-equivariant. Moreover, this decomposed updating procedure prevents the potentially high computational cost incurred by the attention operation while effectively preserving the expressive power. Consequently, the total time complexity of DINES in each round is substantially reduced to $\Theta(NT^2 + N^2)$. In comparison, a self-attention on all player-action pairs incurs $\Theta(N^2T^2)$ time complexity, and the state-of-the-art method (Liu et al., 2024) has a $\Theta(T^{2N})$ complexity due to self-attention operations on all $T^N$ joint action profiles.

LOSS FUNCTION

After $K$ rounds of iteration, the output mixed strategy $\hat{x}$ is generated by the function $\Phi$ similarly as before. Following the prevailing approach in machine learning-based methods solving Nash Equilibria, we utilize the Nash approximation $\mathrm{NashAppr}(\hat{x})$ as the loss function.

## 5 EXPERIMENTS

In this section, we present our experimental results, evaluating the performance of our DINES model on tabular games and polymatrix games. The goal of our empirical study is threefold. Firstly, we compare our model with baseline deep learning methods on tabular games, to demonstrate the superiority of our model in terms of the accuracy-efficiency trade-off. Secondly, we compare the performance of our model under different number of iteration rounds, showing a improved convergence performance compared with classical learning dynamics. Thirdly, we evaluate our model on poly-matrix games with large number of players, highlighting the scalability and adaptivity of our model in large-scale succinct games.

| Model | Tabular Game | | | |
|---|---|---|---|---|
| | N = 2 T = 16 | N = 2 T = 64 | N = 3 T = 8 | N = 3 T = 16 |
| DINES ($K = 15$) | 0.0296 | 0.0618 | 0.0431 | 0.0510 |
| DINES ($K = 30$) | 0.0242 | 0.0509 | 0.0373 | 0.0442 |
| DINES ($K = 60$) | 0.0229 | 0.0501 | 0.0364 | 0.0432 |
| NFG-Transformer | 0.0243 | 0.0502 | 0.0412 | 0.0297 |
| MLP | 0.2422 | 0.1729 | 0.1143 | 0.0713 |

Table 1: Experimental results in tabular games

| Model | Polymatrix Game | | | |
|---|---|---|---|---|
| | N = 3 T = 16 | N = 4 T = 8 | N = 8 T = 4 | N = 20 T = 4 |
| DINES ($K = 15$) | 0.0642 | 0.0589 | 0.0511 | 0.1201 |
| DINES ($K = 30$) | 0.0572 | 0.0515 | 0.0463 | 0.1187 |
| DINES ($K = 60$) | 0.0546 | 0.0497 | 0.0425 | 0.1150 |

Table 2: Experimental results in polymatrix games

### 5.1 DATA GENERATION AND EXPERIMENT SETUP

For tabular games with $N$ players and $T$ actions, the utility functions are denoted as $N$ tables each of size $T^N$. Following existing works, we generate each entry in the table independently from the uniform distribution $U[-1, 1]$.

For polymatrix games with $N$ players and $T$ actions, we generate $N(N-1)$ payoff matrices, each is $T \times T$ with entries independently following $U[-1, 1]$.

The experiments are done on a NVIDIA TITAN V GPU. We select a embedding dimension of $D = 32$ for our model throughout all experiments.

### 5.2 TABULAR GAMES

The experimental results on tabular games is presented in Table 1. The methods in Duan et al. (2023a) and Liu et al. (2024) are taken as baselines.[2] Across all evaluated settings, DINES consistently achieves the highest accuracy or performs comparably to the best. Given the significantly

---

[2]We report that the training process for Liu et al. (2024) using the official code fails to converge in more than $10^6$ seconds. Therefore, the results presented are sourced directly from their published paper.

reduced time complexity, these results demonstrate the superiority of our model in terms of the accuracy-efficiency trade-off.

Comparing the performance of DINES with different number $K$ of total rounds, one can find that the approximation improves as $K$ increases, but not too much. This means selecting $K = 30$ is enough for a good concentration performance. In comparison, traditional learning dynamics require typically $10^5$ rounds of iterations to achieve good approximation in average (Li et al., 2024). This implies the improved concentration performance of our DINES model.

### 5.3 POLYMATRIX GAMES

The experimental results of DINES on polymatrix games of various sizes is presented in Table 2. For existing deep-learning methods, these games are intractable as the utility functions must be represented in tabular form, which has an exponential size of $NT^N$. In contrast, DINES is able to solve these games with affordable running time, and achieves sufficiently good approximations[3]. This clearly demonstrate the scalability of DINES, enabling it to be applied in large scale succinct games.

## 6 CONCLUSION

In this paper, we propose Deep Iterative Nash Equilibrium Solver (DINES), built on a novel framework incorporating deep learning into the iterative structure of classical learning dynamics. Our approach significantly reduce the time complexity to a polynomial level, resulting in enhanced scalability in games with large number of players and actions, especially in large-scale succinct form games. Experimental results on tabular games and polymatrix games demonstrate the superiority of our approach in terms of the accuracy-efficiency trade-off.

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
