# OpenReview forum: "Solving Nash Equilibrium Scalably via Deep-Learning-Augmented Iterative Algorithms"
_ICLR.cc/2025/Conference — Submitted to ICLR 2025_

### Official Review · Reviewer_Zv7E · 2024-10-22

**Soundness:** 1
**Presentation:** 1
**Contribution:** 2
**Rating:** 5
**Confidence:** 3

**Summary:**

This paper explores the application of Deep Learning to learn the NE, and introduces a new neural network architecture.

**Strengths:**

This paper investigates the use of deep learning to learn NE, a novel and promising direction. The authors claim that their proposed network architecture significantly reduces time complexity.

**Weaknesses:**

1. The writing quality of the paper is inadequate. To be honest, when I first read the abstract, I couldn't clearly understand what algorithm the authors are proposing. Even after reading the entire paper, I still don’t have a clear idea of what the authors mean by "iterative algorithms" as mentioned in the abstract.

2. The paper seems too short. I do not find any appendices, and the main content appears to be less than 9 pages. Although ICLR allows papers shorter than 9 pages, most submissions typically utilize 9 to 10 pages for the main body. I am concerned that this paper does not meet the expectations for ICLR.

3. Regarding novelty, the "Related Work" and "Methodology" sections are poorly written. Due to this, I am unable to assess whether the network architecture proposed by the authors is innovative.

4. No runtime comparisons are provided, leaving the effectiveness of the proposed architecture uncertain.

**Questions:**

1. What are the innovations of your work compared to previous works?
2. Could you provide runtime comparisons?

---

> ### Author Response · Authors · 2024-11-25
>
> Thank you for your valuable feedback and constructive comments. Below are our responses to your concerns.
>
> ## Weaknesses
> 1. We apologize for the confusion. We will revise both the abstract and the paper in the future to provide a more explicit description of the iterative algorithm.
> 2. Thank you for your constructive comment. We will provide more detailed description of our motivation, model and experiments in our future version.
> 3. Thank you for your constructive comment. We will revise the Related Work and Methodology sections to improve the clarity.
> 4. Please see Questions.
>
> ## Questions
>
> In the current version of the paper, we mainly compared the computational costs of different methods through asymptotic complexity. Our model achieves exponential improvement in both time and space requirements in terms of the dependence on the number of players $N$ and actions $T$, which significantly enhances the scalability by addressing the curse of dimensionality. We will empirically compare the running time in the future versions.
>
> Here we provide a quantitative discussion. When $N$ is above 100, the inference of existing methods become absolutely infeasible even on hypercomputers, as their time and space complexity are at least $T^N\geq 2^N>10^{30}$. In comparison, our model remains feasible on standard computers in such cases. For example, if we select $K=30$ as number of iterations and $D=100$ as hidden dimensionality, when $N=100,T=100$, our model’s inference requires about $KNT^2D^2\approx 3*10^{11}$ floating point operations, which can be computed in about $300$ seconds by a $10^9$ flops CPU core, or much less time on GPUs.

---

> > ### Comment · Reviewer_Zv7E · 2024-11-26
> >
> > As Reviewer yi82 said, my points on weakness still hold. Hence, I keep my score.

---

### Official Review · Reviewer_bGJt · 2024-10-22

**Soundness:** 1
**Presentation:** 3
**Contribution:** 2
**Rating:** 3
**Confidence:** 4

**Summary:**

The paper describes a deep neural network architecture for computing an approximate NE in normal-form and polymatrix games. The architecture is permutation equivariant and works by querying the payoffs under different play distributions at each layer, resulting in much more efficient computation. Results are reported on normal-form and polymatrix games.

**Strengths:**

The paper describes computing approximate Nash equilibrium in general-sum N-player games which is a known hard problem. Any work that seeks to compute equilibria faster or more efficiently is important and relevant.

Computation is avoided on activations that are of the order of the size of the payoffs of a game. Instead, computation is done on expected payoffs under mixed play, resulting in significantly smaller computational costs.

**Weaknesses:**

Some properties of the network, such as permutation equivariance, and transformer layers are contributions of prior work (Duan, Marris, Liu), and this paper borderline claims these as contributions.

The discussions around complexity are misleading. I would probably avoid using “polynomial time” (see questions).

I was surprised that DINES was able to match NfgTransformers performance, despite only getting to query expected payoffs. I worry that the authors have made an error in their comparison to Nfg resulting in a factor of 3 discrepancy between the reported results (details provided in questions).

The questions below expand on the weaknesses. I will be willing to increase my score if I have misunderstood any part of the paper, the questions I have are answered, and any additions I have requested are included. I hope my review can be used to strengthen your work.

**Questions:**

[Training] The exact training procedure is under-explained. L382 says that NashAppr is minimized. I think the confusion has arisen because the paper introduces the algorithm as an iterative procedure, but I think I am meant to interpret it as a DNN for the purposes of training? I have made some guesses that should be made explicit in the writing:
a) the network is trained over a distribution of games from U[-1,+1]
b) at test time it is evaluated on any game from U[-1,+1]
c) the parameters are trained by backpropping through all the layers.
Questions:
Do you use the same network for different sized games? It seems that the parameters are independent for the size of the game.
I would like to see a training curve. How long does this network take to train?

[Architecture] On the emphasis on “iterative” in the algorithm. Sure, I get the inspiration from iterative methods, but I think the writing over-emphasizes the iterative properties. The model is essentially a DNN with multiple layers, with the layers being iterations. I think the emphasis would make more sense if the weights of each layer were shared, which made me wonder if you explored this?

[Complexity] “Query-based access” “Only KNT utility queries are made”. When I think of a query of a payoff, I would think of reading a single element of a payoff tensor. However line 227 shows that (in a normal-form-game) all N*T^N elements need to be read to calculate the expected payoffs at each layer, which corresponds to TN “queries” as defined in the paper. I find this accounting a little misleading. I appreciate that this sentence “Moreover, as long as the game has succinct representation, each query of utility function can be efficiently computed,...” tempers the claims a bit, but I think more care should be taken not to mislead the reader, particularly in the abstract. I do understand that DINES only has to process much smaller inputs than NfgTransformer and NES. What is the complexity with respect to? Is it number of strategies or joint strategies?

[Permutation Equivariance] Note that NES and Nfgtransformer also incorporate permutation equivariance. I came away from the paper thinkin that permutation equivariance was a contribution. I don’t understand how randomly initializing the initial embeddings preserves permutation equivariance?

[Equilibrium Selection] The question of equilibrium selection should be discussed. There are many, possibly isolated, NE for a game. The method described in this paper finds an approximate one, but does not say anything about which one is being found. This is a known hard problem in the field, and I have no expectation for this paper to solve it, but some discussion on this will strengthen the paper. For example, I imagine the initial embedding will influence the equilibrium that is selected? Was this explored?

[Experimental Results] I noticed that the code for NfgTransformer did not run, so reported results from the NfgTransformer paper were used. The dataset that NfgTransformer was evaluated on is different to the one DINES evaluated on. In particular, Nfg uses a particular distribution of payoffs with variance of 1 for each element, while DINES uses U[-1, +1], which has variance 1/3. Napkin math means that we approximately expect the scale of the payoffs used in DINES to be 1/3 of that in Nfg. Which will result in 1/3 of the expected exploitability. Therefore I believe the paper is underreporting the performance of the competitor model by about 1/3. Aside: I think all these types of papers should report expected exploitability under a uniform distribution for the datasets they sample from to avoid these types of comparison errors. Can you add a row that shows this?

Minor:

L11/L18: “Nash Equilibrium” -> “Nash equilibrium”

L18: “to solve Nash equilibria more efficiently … reduced time complexity to a polynomial level…”. Should change to “solve *approximate* Nash equilibria”. Claiming polynomial for solving Nash is an overreach.

L107: For the learning dynamics section, the authors should clarify that learning dynamics converges to equilibria in time average. While the theoretical and deep learning methods provide a last-iterate solution.

L167: “e > 0”. Perhaps the authors meant “e >= 0”?

L172: “Nash approximation”. I have seen the terms “NashConv”, “NashGap”, and “exploitability” used in the literature. Should another one be introduced?

L190: Polymatrix is also a succinct game. And the paper only looks at dense and polymatrix. I think drop the succinct section?

L203: Marris and Lui (cited elsewhere in the paper) also utilize permutation equivariance.

L231: The capital N has been used for “number of players” up to this point. “n” -> “N”

L266: “allow a centralized update of all inner states in this framework…” I don’t understand this sentence.

L287: “suffers” -> “suffer”

L290: “n” -> “N”

L329: Why not sample a distribution from a flat dirichlet?

L333: “a algorithm” -> “an algorithm”

L392: “poly-matrix” -> “polymatrix”

L400: How many samples were used to evaluate the tabular results?

---

> ### Author Response · Authors · 2024-11-25
>
> Thanks for your valuable review and constructive comments. We will incorporate them to improve this work in the future. Below are our responses to some of your questions:
>
> **Q1** Some properties of the network, such as permutation equivariance, and transformer layers are contributions of prior work (Duan, Marris, Liu), and this paper borderline claims these as contributions.
>
> **A1** We apologize for the confusion. Our intention was to highlight the advantages of our proposed model, including the permutation equivariance property and the expressiveness from transformer structure, instead of claiming the two concepts as our contributions. Still when incorporating them into our model, we make improvements compared with prior works: We generalize the permutation equivariance concept to randomized output distribution, to avoid restriction on equilibrium selection, which may harm the social welfare. We design an efficient attention structure with $O(NT^2+N^2)$ time complexity, while NfgTransformer takes $O(T^{2N})$ time since it performs self-attention operation on all $T^N$ joint actions.
>
> **Q2** I worry that the authors have made an error in their comparison to Nfg resulting in a factor of 3 discrepancy between the reported results.
>
> [Experimental Results] I noticed that the code for NfgTransformer did not run, so reported results from the NfgTransformer paper were used. The dataset that NfgTransformer was evaluated on is different to the one DINES evaluated on. In particular, Nfg uses a particular distribution of payoffs with variance of 1 for each element, while DINES uses U[-1, +1], which has variance 1/3. Napkin math means that we approximately expect the scale of the payoffs used in DINES to be 1/3 of that in Nfg. Which will result in 1/3 of the expected exploitability. Therefore I believe the paper is underreporting the performance of the competitor model by about 1/3. Aside: I think all these types of papers should report expected exploitability under a uniform distribution for the datasets they sample from to avoid these types of comparison errors. Can you add a row that shows this?
>
> **A2** Thank you for your insightful comment. We apologize for the oversight. You are correct that the discrepancy arises from the different payoff distributions used in the NfgTransformer and DINES models. Comparing the variance of payoff distribution, The scale of the payoffs for DINES is approximately $\frac{1}{\sqrt{3}}\approx 0.58$ of that for NfgTransformer, so the loss of NfgTransformer should be scaled with factor $\sqrt{\frac{1}{3}}$.
> We appreciate your suggestion to align the datasets by the exploitability of a uniform strategy. We will take this into consideration in future versions of the paper and will incorporate it in future versions of our paper. We will also continue to refine the accuracy of DINES in future work.
>
> **Q3** [Training] The exact training procedure is under-explained. L382 says that NashAppr is minimized. I think the confusion has arisen because the paper introduces the algorithm as an iterative procedure, but I think I am meant to interpret it as a DNN for the purposes of training? I have made some guesses that should be made explicit in the writing: a) the network is trained over a distribution of games from U[-1,+1] b) at test time it is evaluated on any game from U[-1,+1] c) the parameters are trained by backpropping through all the layers. Questions: Do you use the same network for different sized games? It seems that the parameters are independent for the size of the game. I would like to see a training curve. How long does this network take to train?
>
> **A3** You are right that our model is essentially a single DNN in training, which consists of $K$ networks with the same structure but different parameters.
> a) b) The model is both trained and tested over the distribution of normal form games with payoff distribution U[-1,+1].
> c) You are right that the model is trained by backpropping through all the $K$ networks.
> Thank you for your questions, we will present more detailed descriptions about our model and experiment in the future versions. Currently we are using a fixed model size for different sized games, but the parameters are independently trained for different game distributions.
>
> **Q4** [Architecture] On the emphasis on “iterative” in the algorithm. Sure, I get the inspiration from iterative methods, but I think the writing over-emphasizes the iterative properties. The model is essentially a DNN with multiple layers, with the layers being iterations. I think the emphasis would make more sense if the weights of each layer were shared, which made me wonder if you explored this?
>
> **A4** Thank you for your thoughtful comment. We have tried sharing the network parameters across all iterations, but the initial results are not very promising. However, we recognize the potential of this approach and may revisit it in future work.

---

> > ### Author Response · Authors · 2024-11-25
> > **(continued)**
> >
> > **Q5** [Complexity] “Query-based access” “Only KNT utility queries are made”. When I think of a query of a payoff, I would think of reading a single element of a payoff tensor. However line 227 shows that (in a normal-form-game) all N*T^N elements need to be read to calculate the expected payoffs at each layer, which corresponds to TN “queries” as defined in the paper. I find this accounting a little misleading. I appreciate that this sentence “Moreover, as long as the game has succinct representation, each query of utility function can be efficiently computed,...” tempers the claims a bit, but I think more care should be taken not to mislead the reader, particularly in the abstract. I do understand that DINES only has to process much smaller inputs than NfgTransformer and NES. What is the complexity with respect to? Is it number of strategies or joint strategies?
> >
> > **A5** You are right that computing the utility function for a mixed strategy profile is nontrivial. In our paper, we are actually assuming that each query of a player’s expected utility under a mixed strategy profile takes unit or constant time when calculating the asymptotic time complexity. To justify this, we note that an unbiased estimation can be computed in $O(NT)$ time. Given the constant time access to the utility functions of pure action profiles, a player’s expected utility can be unbiasedly estimated by simply sampling an action for each player following the mixed strategy, which takes $O(NT)$ time. By performing $O(\frac{1}{\epsilon^2})$ times of independent sampling and taking the average, we can obtain an estimation with $O(\epsilon)$ expected error.
> >
> > On the other hand, precise computation of the utility function under a mixed strategy profile indeed takes $\Omega(T^N)$ time in tabular games, as it can depend on every entry of the utility table. Currently we are actually using bruteforce tensor production with exactly this complexity in the training and testing process.
> >
> > Thank you again for pointing this out. We may further investigate this issue in our future work, for example assuming only the access to an unbiased estimator of the utility function, so that a single sample from the mixed strategy profile suffices.
> >
> > **Q6** [Equilibrium Selection] The question of equilibrium selection should be discussed. There are many, possibly isolated, NE for a game. The method described in this paper finds an approximate one, but does not say anything about which one is being found. This is a known hard problem in the field, and I have no expectation for this paper to solve it, but some discussion on this will strengthen the paper. For example, I imagine the initial embedding will influence the equilibrium that is selected? Was this explored?
> >
> > **A6** Thank you for your insightful suggestion. Although we have not investigated the equilibrium selection issues, we may take it into consideration in the future versions, possibly comparing the social welfare of the equilibrium.
> >
> > We also appreciate your detailed minor comments, which are helpful for improving our paper. Here we omit the responses to them as the details may change in the future version of our paper.

---

> ### Comment · Reviewer_bGJt · 2024-11-27
>
> I have read and appreciate the replies of the authors. Unfortunately, I still think the paper is not strong enough in its current form to be accepted at ICLR, so I will not be changing my recommendation.

---

### Official Review · Reviewer_yi82 · 2024-10-29

**Soundness:** 2
**Presentation:** 1
**Contribution:** 2
**Rating:** 3
**Confidence:** 4

**Summary:**

This paper investigates deep-learning-based Nash equilibrium learning algorithms. It proposes an algorithm called Deep Iterative Nash Equilibrium Solver (DINES) by leveraging query-based access to utility functions rather than requiring the full utility matrix.

**Strengths:**

This paper proposes a new deep-learning based Nash equilibrium learning algorithm, which outperforms previous algorithms with lower computational overhead.

**Weaknesses:**

1.  The presentation should be improved.

                  - The authors should include a figure to clearly illustrate the differences between their network architecture and that of previous works.

                  - More details about the proposed algorithm need to be provided. In the current version, both Algorithm 1 and Algorithm 2 lack critical information, such as how the network parameters are updated.

                  - The authors claim "improved convergence," which I believe is inaccurate. Proving theoretical convergence for deep learning algorithms is notoriously challenging.

                  - The abstract fails to highlight the paper's contributions. In fact, I would suggest a complete rewrite of the abstract to more clearly emphasize the key innovations of this work.

2. Apart from the issue of representation, the convergence of the proposed algorithm also raises concerns. It is unclear whether the proposed algorithm can guarantee convergence to a Nash equilibrium.

 3. The novelty seems limited because the proposed approach seems to only modify the method to access the utility matrix in the original transformer-based algorithm.

4. The significance of this paper seems limited. By leveraging query-based access to utility functions rather than requiring the full utility matrix, it seems that the proposed method only reduces the computational overhead in sparse games.  Succinct games and other sparse games inherently require less space for representation.

5. Experimental results are not convincing. In experiments, we can see that the revised transformer-based algorithm cannot significantly outperform the original transformer-based algorithm in most tabular games. The time should be shown as well.
In Polymatrix games, I believe the original transformer-based algorithm can still output results for small games. For example, the result for N=3 and T =16 has been shown in tabular games and should be shown in Polynamtrix games. In addition, the game with N=3 and T=16 and the game with N=4 and T=8 should have similar time complexity for the original transformer-based algorithm.

**Questions:**

see the weakness

---

> ### Author Response · Authors · 2024-11-25
>
> Thanks for your constructive comments and suggestions, we will incorporate them to improve our paper. Below are our responses to some of your questions.
> 1. Thanks for your valuable suggestions, we will present more detailed explanation of network architecture with figures in our future version.
> In the current version of the abstract, we primarily emphasized the exponential improvement in the computational efficiency achieved by our algorithm, which significantly enhances the scalability by addressing the curse of dimensionality. We will thoroughly revise it to improve the clarity.
> 2. You are right that it is unlikely to theoretically guarantee convergence for any efficient algorithm, given the PPAD-hardness of Nash equilibrium computation. The claim of “improved convergence” is empirical, supported by the fact that the number of iterations is improved from $10^5$ to $30$ compared with traditional learning dynamics, while achieving improved accuracy.
> 3. Compared to the baselines, especially (Liu et al, 2024), our novelty is threefold:
>     1. Our model takes polynomial number of query accesses to the utility functions, instead of inputting the whole utility table. This reduces the asymptotic running time from exponential to polynomial in the number of players and actions, and allows our framework to be applied in any normal-form games as long as the utility function can be efficiently computed.
>    2. In terms of the model structure, compared to (Liu et al, 2024), we design a decomposed attention operation among player embeddings and action embeddings, which also contributes to the exponential improvement of time efficiency, as the transformer operation is the bottleneck of time cost.
>    3. Our model produces an updated strategy profile in every iteration, which can be interpreted as a process approaching equilibrium through repeated interaction. This approach combines adaptability and interpretability, similar to traditional learning dynamics, which are lacking in previous learning-based methods.
> 4. Significance: Our model achieves exponential improvement of both time and space efficiency in terms of the dependence on the number of players $N$ and actions $T$, even for succinct games. Existing deep-learning methods require at least $\Theta(NT^N)$ memory usage even for succinct games, since they must compute the whole utility table. This also results in a $\Theta(NT^N)$ time usage, and $\Theta(T^{2N})$ for (Liu et al., 2024)’s model. In contrast, given that the utility function is efficently computable, our model requires only $\Theta(NT)$ space and $\Theta(N^2+NT^2)$ time.
> Existing methods become absolutely infeasible even on hypercomputers when $N$ is above 50 or 100, while our model remains feasible on standard CPUs. For example, if we select $K=30$ as number of iterations and $D=100$ as hidden dimensionality, when $N=100,T=100$, our model’s inference can be computed in an estimated time of $300$ seconds by a $10^9$ flops CPU core, or much less time on GPUs.
> 5. Thanks for your constructive suggestions, we will incorporate them and present more experimental details in our future version.

---

> > ### Comment · Reviewer_yi82 · 2024-11-26
> >
> > Thank you for your response. I think my points on weakness still hold.

---

### Official Review · Reviewer_7p4p · 2024-11-01

**Soundness:** 2
**Presentation:** 4
**Contribution:** 2
**Rating:** 3
**Confidence:** 4

**Summary:**

The paper proposes a deep-learning based approach that iteratively approximates Nash equilibrium in normal-form games. The proposed approach shares parameters for all players, and optimizes the parameters by backproping on the Nash approximation loss. It achieves comparable performance as existing deep-learning baselines while scaling better to the number of players and the size of action space.

**Strengths:**

The writing and presentation is very good. The authors clearly motivates the problem and covers necessary background. The proposed framework is also well motivated and clearly explained. I like that the authors provided lots of intuition behind the method, making the presentation very smooth.

**Weaknesses:**

- The main weakness is that the experiments are not thorough enough. A central claim is that the proposed DINE framework is more computationally efficient. In order to back up that claim, I would expect to see strong empirical evidence. However, the only experiments are relatively small-scale (except for maybe the polymatrix game with larger N) with utilities generated from uniform distribution [-1, 1]. I don’t think those games are representative enough. Sometimes it appears that the baselines are better (e.g. NFG-transformer has better performance in both the score and the number of iterations in tabular games). For polymatrix games, there are no baseline at all, as the authors claim that they are intractable for existing deep-learning methods. However, in (Liu et al, 2024), they reported performance on polymatrix games, and it’s probably a good idea to compare to that. For Table 2, at least for smaller N values, you can report the baseline performance as well.
- By using i.i.d. stochastic initialization, I believe the permutation equivariance is only achieved in distribution, but not exact for a given instance (a specific learned strategy), it’s not guaranteed to have permutation equivariance. In contrast, (Liu et al, 2024) have exact permutation equivariance.

Siqi Liu, Luke Marris, Georgios Piliouras, Ian Gemp, and Nicolas Heess. Nfgtransformer: Equiv-
ariant representation learning for normal-form games. CoRR, abs/2402.08393, 2024. doi: 10.
48550/ARXIV.2402.08393. URL https://doi.org/10.48550/arXiv.2402.08393.

**Questions:**

- I’m confused about what K or the number of iteration means in this paper. From Algorithm 2, it appears that K refers to the number of steps it takes for a **learned** network to generate a reasonable strategy profile. However, in your experiments (line 437), you are comparing it to traditional methods that require typically 10^5 rounds of iterations. The cited source (Li et al. 2024) is a survey paper, and the relevant sentence is “Learning dynamics: We conduct T = 105 iterations with zero initialization.” It’s unclear which method is associated with that number of iterations, or whether “iteration” means the same thing as in this paper. Also, the baseline NFG-Transformer (Liu et al, 2024) has K=2-8 in their experiment, which is even smaller than the K=~30 in this paper. It seems against the claim that DINES model has superior convergence.
- Training details are not discussed in this paper. Can you report training curves?
- Please explain the metric used in experiments (e.g. what are the numbers in Table 1 and 2?)
- Since you mentioned that assymmetric actions are important to achieve good loss social welfare, can you show experimental evidence for that? I.e. compare the loss for identical and random initialization.
- One of the claimed advantages of the proposed method is “structural explainability”. I don’t quite get how the resulted algorithm can be interpreted as “learning dynamics”. Can you clarify what learning dynamics means in this context, and how do you deduce that from the resulted algorithm?
- Line 320: “Recording the last mixed strategy allows a residual updating of strategies, which is described later”. I couldn’t find the description on this later in the paper. Could you clarify how the strategy is updated in a residual way?

Hanyu Li, Wenhan Huang, Zhijian Duan, David Henry Mguni, Kun Shao, Jun Wang, and Xiaotie
Deng. A survey on algorithms for nash equilibria in finite normal-form games. Comput. Sci.
Rev., 51(C), June 2024. ISSN 1574-0137. doi: 10.1016/j.cosrev.2023.100613. URL https:
//doi.org/10.1016/j.cosrev.2023.100613.

Siqi Liu, Luke Marris, Georgios Piliouras, Ian Gemp, and Nicolas Heess. Nfgtransformer: Equiv-
ariant representation learning for normal-form games. CoRR, abs/2402.08393, 2024. doi: 10.
48550/ARXIV.2402.08393. URL https://doi.org/10.48550/arXiv.2402.08393.

---

> ### Author Response · Authors · 2024-11-25
>
> Thanks for your thorough investigation and constructive comments on our paper. We will incorporate your comments to improve the future versions of our paper, especially the sufficiency of experiments. Below are our responses to your questions.
> ## Weaknesses
> 1. We appreciate your constructive suggestions about the experiment, we will incorporate them and present more comprehensive experiment results in the future version of our paper.
> 2. You are right that we have generalized the concept of permutation equivariance to the output distribution. Our purpose is to avoid the restriction of equilibrium selection posed by permutation equivariance (which may significantly harm social welfare, as mentioned in Duan et al, 2023b), while taking advantage of the inherent symmetricity of the problem.
> ## Questions
> 1. We apologize for the confusion. A round means the algorithm queries the utility under a strategy profile and generate an updated strategy profile accordingly. For the traditional learning dynamics, a common implementation is to set a fixed number K of rounds instead of repeating the iteration until convergence, and typically K=100000. Our model’s hyper parameter K similarly indicates the number of rounds, and we actually learn K networks, representing each round’s updating operation of strategies and inner states. By experiments, we find that taking K=30 is enough for good accuracy, meaning that the updating rule learned by our model can recover equilibrium more efficiently than traditional methods. It is true that Liu et al. only require K=2-8 iterations, but their model accesses the whole utility table in each iteration, and performs attention operation on all pure strategy profiles, so their computational cost in each iteration is exponentially larger than ours. Therefore, our algorithm have a significant advantage in efficiency.
> 2. Thank you for the suggestion, we will present training curves in future versions.
> 3. The metric of accuracy is Nash Approximation defined in section 2.
> 4. Thank you for your suggestion, we will present comparison on social welfare in future versions.
> 5. Our proposed framework (Algorithm 2) follows the common interaction-and-update structure of traditional learning dynamics (Algorithm 1), where players repeatedly update their strategy based on the utility in the interaction with all other players in each round. This is motivated by the heuristic idea that players will reach Nash equilibrium through repeated interactions in many scenarios. We can expect the model to learn the updating rules no worse than the tradtional learning dynamics during training, which provides an evidence for its reliability .
> 6. We apologize for missing the description of the residual update of strategy. In each round, we actually compute
> $\mathbf{x}\_p^{(k)}=(1-\gamma\_p^{(k)})\mathbf{x}\_p^{(k-1)}+\gamma_p^{(k)}\Phi(y\_{p}^{(k-1)};\theta\_{\Phi}^{(k)})$
> where $\gamma\_p^{(k)}$ is an adaptive ratio calculated from the player embedding $\beta\_{p}^{(k-1)}$ by an FFN.

---

### Meta-Review · Area_Chair_XXQE · 2024-12-23

**Metareview:**

This paper attempts to design deep-learning-augmented iterative algorithm to find Nash equilibrium efficiently especially when the number of player is large. While such problem is PPAD-hard in the worst case in the theoretical, this paper attempts to design empirical algorithms that performs well in practice. There are multiple concerns regarding the clarity of the paper, the thoroughness of the experiments, and the significance of the overall content for the current version of the paper. We therefore recommend rejection.

**Additional Comments On Reviewer Discussion:**

All reviewers unanimously recommend rejection.

---

### Decision · Program_Chairs · 2025-01-22

Reject